



# Flood trends in Europe: are changes in small and big floods different?

Miriam Bertola[1], Alberto Viglione[2], Julia Hall[1], and Günter Blöschl[1]

[1]Institute of Hydraulic Engineering and Water Resources Management, Vienna University of Technology, Karlsplatz 13, 1040, Vienna, Austria

[2]Department of Environmental Engineering, Land and Infrastructure, Polytechnic University of Turin, Corso Duca degli Abruzzi 24, 10129 Torino, Italy

**Correspondence:** Miriam Bertola (bertola@hydro.tuwien.ac.at)

**Abstract.**

Recent studies have revealed evidence of trends in the median or mean flood discharge in Europe over the last five decades, with clear and coherent regional patterns. The aim of this study is to assess whether trends also occurred for larger return periods accounting for the effect of catchment scale. We analyze 2370 flood records, selected from a newly-available pan-European flood database, with record length of at least 40 years over the period 1960-2010 and with contributing catchment area ranging from 5 to 100 000 km$^2$. To estimate regional flood trends, we use a non-stationary regional flood frequency approach consisting of a regional Gumbel distribution, whose median and growth factor can vary in time with different strengths for different catchment sizes. A Bayesian Monte Carlo Markov Chain (MCMC) approach is used for parameter estimation. We quantify regional trends (and the related sample uncertainties), for floods of selected return periods and for selected catchment areas, across Europe and for three regions where coherent flood trends have been identified in previous studies. Results show that, in the Atlantic region, the trends in flood magnitude are generally positive. In small catchments (up to 100 km$^2$), the 100-year flood increases more than the median flood, while the opposite is observed in medium and large catchments, where even some negative trends appear, especially over the southern part of the Atlantic region. In the Mediterranean region flood trends are generally negative. The 100-year flood decreases less than the median flood and, in the small catchments, the median flood decreases less compared to the large catchments. Over Eastern Europe the regional trends are negative and do not depend on the return period, but catchment area plays a substantial role: the larger the catchment, the more negative the trend. The process causalities on the effects of return period and catchment area on the flood trends are discussed.

## 1 Introduction

Increasing flood hazard in Europe has become a major concern as a consequence of severe flood events experienced in the last decades. Examples of such events are the extreme floods occurred in central Europe in 2002 (e.g. Ulbrich et al., 2003) and 2013 (e.g. Blöschl et al., 2013a), and in northwest England the winter floods of 2009 (e.g. Miller et al., 2013) and 2015-16 (e.g. Barker et al., 2016). Hence a growing number of flood trend detection studies has been published in recent years. These studies typically analyse a large set of time series of flood peaks in a region and test them for the presence of significant gradual or



abrupt changes in flood magnitude or frequency. For example, Petrow and Merz (2009) analysed eight flood indicators, from
145 gauges in Germany over the period 1951-2002, and detected mainly positive trends in the magnitude and frequency of
floods. Villarini et al. (2011) tested the flood time series of 55 stations in central Europe, with at least 75 years of data, for
abrupt or gradual changes and found mostly abrupt changes associated with anthropogenic intervention. Mediero et al. (2014)
detected a general decreasing trend in the magnitude and frequency of floods in Spain, with the exception of the north-west.
Prosdocimi et al. (2014) investigated the presence of trends in annual and seasonal maxima of flow peaks in the UK and
found clusters of increasing trends for winter peaks in northern England and Scotland, and decreasing trends for summer peaks
in southern England. These studies are highly heterogeneous in terms of flood data types, period of records and detection
approaches and it is therefore not trivial to deduce regional patterns of flood regime change at the larger continental scale.
Despite this fragmentation, Hall et al. (2014) summarized their findings in a qualitative map of increasing, decreasing and not
detectable flood changes for Europe, where consistent regional patterns emerge. In particular, over central and western Europe
flood magnitude appears to increase with time, while it appears to decrease over the Mediterranean catchments and in eastern
Europe.

More recently, thanks to the availability of European and global high spatial resolution databases, large-scale investigation
studies across Europe have been published. Mangini et al. (2018) extracted 629 flood records, from the Global Runoff Data
Center database (GRDC, 2016), and compared the detected trends in magnitude and frequency of floods from different ap-
proaches (annual maximum flood and peak over threshold) for the period 1965–2005. Blöschl et al. (2019) analysed 2370
flood records from a newly available pan-European flood database, consisting of more than 7000 observational hydrometric
stations and covering the last five decades (Hall et al., 2015), and revealed consistent spatial patterns of trends in the magnitude
of the annual maximum flood, with clear positive trends over the Atlantic regions and decreasing trends over eastern Europe
and the Mediterranean.

Most of the existing studies typically analyse catchments individually and investigate whether spatial clusters or coherent
regional patterns of flood trends can be observed (e.g. Petrow and Merz, 2009; Prosdocimi et al., 2014; Mangini et al., 2018).
Based on predefined regions or obtained change patterns, some studies aggregate flood records and local test results in order
to assess their field significance (e.g. Douglas et al., 2000; Mediero et al., 2014; Renard et al., 2008). The main limitation
of most of the at-site studies is the limited length of the flood peak records locally available for the detection of trends,
resulting in low signal-to-noise ratio and hence high uncertainties in the detected trend. Increasing the signal-to-noise ratio
can be achieved by pooling flood data over multiple sites within a homogeneous region, as in regional frequency analyses
(Dalrymple, 1960; Hosking and Wallis, 1997). Several studies propose non-stationary regional frequency analyses for changes
in precipitation extremes and flood trends, that consider the dependency of the regional estimates on time (e.g. Cunderlik
and Burn, 2003; Renard et al., 2006a; Leclerc and Ouarda, 2007; Hanel et al., 2009; Roth et al., 2012) or on climatic and
anthropogenic covariates (e.g. Lima and Lall, 2010; Tramblay et al., 2013; Renard and Lall, 2014; Sun et al., 2014; Prosdocimi
et al., 2015; Viglione et al., 2016). Other approaches analyse coherent regional change by testing the presence of trends in
regional variables, as the number of annual floods in the region (e.g. Hannaford et al., 2013), or with regional tests (e.g.
Douglas et al., 2000; Renard et al., 2008).





Most of the above cited studies however investigate changes in the mean annual (or median) flood only, and few examples
exist where observed trends in different flood quantiles are analysed. Typically, flood quantiles obtained with stationary and
non-stationary flood frequency approaches are compared (see e.g. Machado et al., 2015; Šraj et al., 2016; Silva et al., 2017).
The detection of changes in the magnitude of flood quantiles is much more common for precipitation (e.g. Hanel et al., 2009)
or in flood projection studies (e.g. Prudhomme et al., 2003; Leander et al., 2008; Rojas et al., 2012; Alfieri et al., 2015).

To address this research gap, the aim of this study is to assess the changes occurred in small vs. big flood events (corre-
sponding to selected flood quantiles) across Europe in the last five decades, and to determine whether these changes have been
subjected different degrees of modification in time. Moreover, given that the impacts of different drivers of change on floods
are expected to be strongly dependent on spatial scales (Blöschl et al., 2007; Hall et al., 2014), it is here also of interest to
assess the effect of catchment area, by comparing changes of flood quantiles in small and large catchments. Since the length of
at-site flood records is often not sufficient to enable flood quantiles associated with high return periods (i.e. low probability of
exceedance, e.g. the 100-year flood) to be reliably estimated, we adopt in this study a (non-stationary) regional flood frequency
approach, which pools flood data of multiple sites in order to increase the robustness of the estimated regional flood frequency
curve with its changes over time. The methods and the flood database are described in detail in Sect. 2. The results are presented
in Sect. 3, where we show the estimation of the flood quantiles and their trends in one example region (Sect. 3.1), the patterns
of flood regime change emerging from a spatial moving window analysis over Europe (Sect. 3.2) and the flood regime changes
in three relevant regions, emerging from the change patterns (Sect. 3.3).

## 2  Methods

### 2.1  Regional flood change model

We aim to quantify the changes in time in the flood frequency curve by calculating the relative change in time of flood quantiles
corresponding to different return periods for catchments of different size. To this aim, we propose a regional flood change model
that is more robust than local (at-site) trend analysis, in particular regarding trends associated to large quantiles of the flood
frequency curve. We assume the flood peaks to follow a Gumbel distribution, whose cumulative distribution is defined as:

$$F_X(x) = p = \exp\left(-\exp\left(-\frac{x-\xi}{\sigma}\right)\right) = \exp\left(-\exp\left(-y\right)\right)$$

where $\xi$ and $\sigma$ are the location and scale parameter and

$$y = \frac{x-\xi}{\sigma} = -\ln(-\ln p)$$

is the Gumbel reduced variate. The corresponding quantile function, i.e., the inverse of the cumulative distribution function, is:

$$q(p) = \xi - \sigma \ln\left(-\ln p\right) = \xi + \sigma y$$

In this paper we consider two alternative parameters, which better relate to the literature on regional frequency analysis,
specially to the Index-Flood method of Dalrymple (1960) and Hosking and Wallis (1997). The alternative parameters are: (1)





the 2-years quantile or median $q_2$ (which corresponds to the index-flood), and (2) the 100-yr growth factor $x'_{100}$, which gives
the 100-years quantile as $q_{100} = q_2(1 + x'_{100})$ in a similar fashion to the modified quantiles in Coles and Tawn (1996) and
Renard et al. (2006b). The relationships of these alternative parameters with the Gumbel location and scale parameters are:

$$q_2 = \xi + \sigma y_2$$

$$x'_{100} = \sigma(y_{100} - y_2)/(\xi + \sigma y_2)$$

where $y_2 = -\ln(-\ln(0.5))$ and $(y_{100} - y_2) = -\ln(-\ln(0.99)) + \ln(-\ln(0.5))$. The quantile function, with the alternative
parametrisation, is here expressed as a function of the return period $T$ as:

$$q_T = q_2 \left(1 + a_T x'_{100}\right) \tag{1}$$

where $a_T = (y_T - y_2)/(y_{100} - y_2)$ and $y_T = -\ln(-\ln(1 - 1/T))$. In particular, $a_T$=0 for $T$=2 and $a_T$=1 for $T$=100.

In the following we estimate the parameters of the Gumbel distribution both locally and regionally. For the local case, we
allow the parameters to change with time according to the following log-linear relationships:

$$\ln q_2 = \ln \alpha_{2_0} + \alpha_{2_1} \cdot t$$

$$\ln x'_{100} = \ln \alpha_{g_0} + \alpha_{g_1} \cdot t$$

For the regional case we introduce the scaling with catchment area of the parameters, according to the following relationships:

$$\ln q_2 = \ln \alpha_{2_0} + \gamma_{2_0} \ln S + (\alpha_{2_1} + \gamma_{2_1} \ln S) \cdot t + \varepsilon$$

$$\ln x'_{100} = \ln \alpha_{g_0} + \gamma_{g_0} \ln S + (\alpha_{g_1} + \gamma_{g_1} \ln S) \cdot t$$

$$\varepsilon \sim \mathcal{N}(0, \sigma)$$

where the $\varepsilon$ term accounts for the fact that additional local variability, on top of the one explained by time and catchment
area, is affecting the index flood but not the growth curve. In our model, a homogeneous region is thus formed by sites whose
growth curve depends on catchment area and time only and whose index flood also depends on other factors which determine
an additional noise (here assumed normal). Spatial cross-correlation between flood timeseries at different sites is not accounted
for in this study.

In order to quantify the changes in time in the flood frequency curve, we calculate the relative change in time of the generic
flood quantile $q_T$ (defined in Eq. 1), which for the local case is:

$$\frac{1}{q_T} \frac{dq_T}{dt} = \alpha_{2_1} + \alpha_{g_1} - \frac{\alpha_{g_1}}{1 + a_T x'_{100}} \tag{2}$$

and for the regional case is:

$$\frac{1}{q_T} \frac{dq_T}{dt} = \alpha_{2_1} + \alpha_{g_1} + (\gamma_{2_1} + \gamma_{g_1}) \ln S - \frac{\alpha_{g_1} + \gamma_{g_1} \ln S}{1 + a_T x'_{100}} \tag{3}$$

The alternative parameters, the quantiles and their local and regional relative trends are estimated by fitting the local and
regional models to flood data with Bayesian inference through a Markov Chain Monte Carlo approach. One of the advantages of





the Bayesian MCMC approach is that the credible bounds of the distribution parameters (and other estimated quantities) can be directly obtained in the estimation procedure without any additional assumption. The R package *rStan* (Carpenter et al., 2017)

is used to perform the MCMC inference. It makes use of Hamiltonian Monte Carlo sampling, which speeds up convergence and parameter exploration by using the gradient of the log posterior (Stan Development Team, 2018). For each inference, we generate 4 chains of length $N_{sim}$=100 000, each starting from different parameter values, and check for their convergence. An improper uniform prior distribution over the entire real line is set for the parameters, with the exception of $\alpha_{2_0}$ and $\alpha_{g_0}$ for which we use an improper uniform prior distribution over the entire positive real line. When fitting the regional model we make

the assumption of regional homogeneity with regards to the distribution of flood peaks, allowing local variability of the median value and its changes in time.

## 2.2 European flood database

In this study we analyse annual maximum discharge series from a newly available pan-European flood database, consisting of more than 7000 observational hydrometric stations and covering the last five decades (Hall et al., 2015). Their contributing

catchment areas range from 5 to 100 000 km$^2$. The flood discharge data are accessible at https://github.com/tuwhydro/europe_floods.

Only the stations satisfying the following selection criteria, based on record length and even spatial distribution, are considered for the estimation of the regional trends. As in Blöschl et al. (2019), we select stations with at least 40 years of data in the period 1960-2010, with record starting in 1968 or earlier, and ending in 2002 or later. Additionally, in order to ensure a more

even spatial distribution across Europe, in Austria, Germany and Switzerland (countries with highest density of stations in the database) the minimum record length accepted is 49 years, in Cyprus, Italy and Turkey 30 years and in Spain 40 years without restrictions to the start and end of the record. Figure 1 shows the locations of the 2370 station satisfying the above selection criteria.

## 2.3 Experimental design for the regional analyses

To assess the regional trends in small and large floods and in small to large catchments across Europe, we fit the regional flood change model, described in Sect. 2.1, to overlapping spatial windows of dimension 600x600 km, assumed to be homogeneous. The overlapping length is 200 km in both directions. The size and overlapping length of the windows is chosen, after several tests, in order to ensure a sufficient number of gauges within each window and an appropriate spatial resolution at which to present regional trends at the continental scale. Significant differences in regime changes, when changing the window size,

are not observed (not shown). As the focus of the study is on the overall flood regime changes at the large scale across the continent, the selected 600x600 km windows can be considered, in this context, homogeneous with regards to geographical location and hence regional climate, which is the most important factor influencing the timing of annual maximum floods in Europe (Hall and Blöschl, 2018).

Figure 1 shows the resulting 200x200 km grid cells. Each of the 600x600 km windows considered in the analysis is com-

posed of 9 neighbouring cells as represented, for example, by the black rectangular region, whose regional trend estimates are





analysed in detail in Sect. 3.1. The example region is selected over central Europe because of the density of available gauges with different ranges of contributing catchment areas. In each window we estimate the regional relative trend in time of $q_2$ and $q_{100}$ (i.e. percentage change of the 2-year and 100-year floods), as defined in Eq. 3, for small and big catchments (i.e. assuming $S$=100 and 10 000 km²). Note that this analysis intends to show the estimated flood trends in hypothetical catchments with

a specific size, which do not exist everywhere across Europe, based on fitting the model to existing catchments. We plot the resulting trends on a map by assigning their values to the central cell of the window (e.g. the light red area in Fig. 1).

Additionally, Fig. 1 shows three regions (numbers 1-3) that were identified by Blöschl et al. (2019) based on spatial patterns of flood trends and distinct driving processes. These regions are not dissimilar, in terms of size and geographic location, to three of the European sub-domains usually considered in climate modeling studies, which represent comparatively homogeneous

climatic conditions (Kotlarski et al., 2014). Table 1 shows some related regional summary statistics. We also fit the regional change model to each of these three regions and trends in small and big floods for small to large catchments are analysed (Sect. 3.3).

In summary, the following regional analyses are carried out:

- In Sect. 3.1 regional flood regime changes over central Europe are investigated. The regional model is fitted to the black

rectangular region of Fig. 1, taken as an example and containing 601 hydrometric stations. The regional model flood quantiles and their trends in time are shown, for this region, as a function of catchment area and of return period (as defined in Eq. 1 and Eq. 3, respectively). The regional trends in $q_2$ and $q_{100}$ are finally compared for five hypothetical catchment sizes (S=10, 100, 1000, 10 000 and 100 000 km²). Additionally, local trend estimates (as in Eq. 2) are shown together with the regional trends.

- In Sect. 3.2 regional flood regime changes across Europe are investigated. The regional model is fitted to overlapping windows across Europe, of size 600x600 km, and the regional trends in $q_2$ and $q_{100}$ are estimated for small and big hypothetical catchments (S=100 and 10 000 km², respectively). Maps of the estimated trends are shown, where the trend values are plotted in the respective central 200x200 km cell of each region. Differences among the estimated trends across Europe are additionally calculated for further comparison.

- In Sect. 3.3 regional flood regime changes in the Atlantic, Mediterranean and eastern European regions are investigated. The regional model is fitted to the three regions (1-3) of Fig. 1, resulting from the change patterns, and the regional trends in $q_2$ and $q_{100}$ are estimated and compared for five hypothetical catchment sizes (S=10, 100, 1000, 10 000 and 100 000 km²).



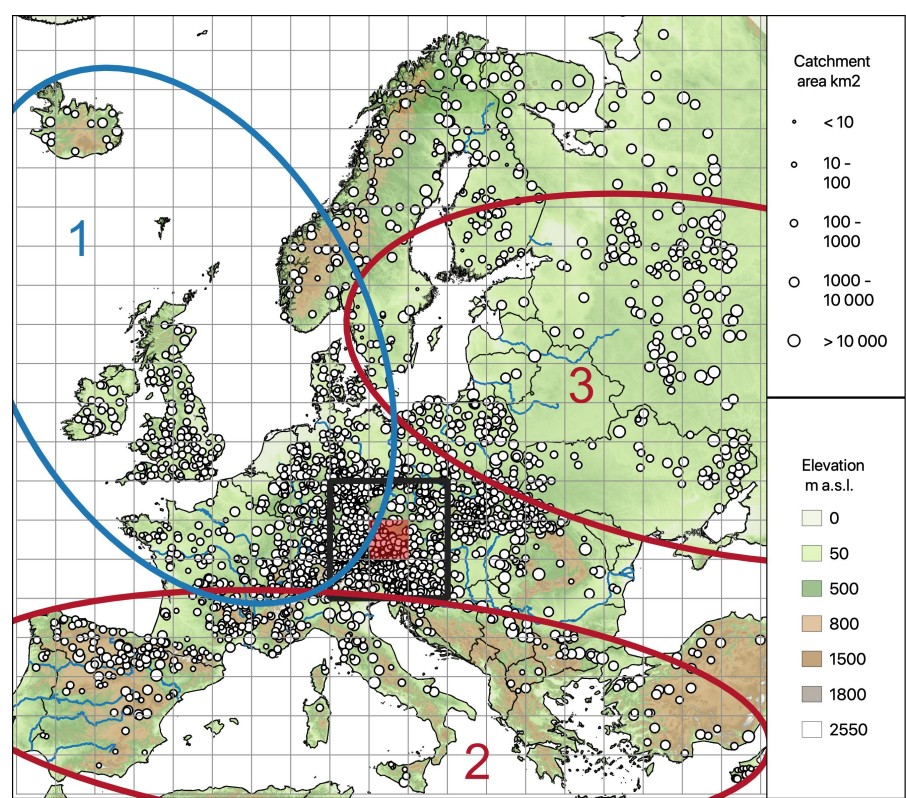

**Figure 1.** Map of the locations of the selected 2370 hydrometric stations in Europe and regions considered in this study. The size of the circles is representative for the contributing catchment area. The size of the grid cells is 200x200 km. The black rectangle shows the size of the spatial moving windows analysed in Sect. 3.2. It consists of 6 cells, corresponding to 600x600 km. The three ellipses (numbers 1-3) mark homogeneous sub-regions concerning trends, analysed in Sect. 3.3, and consist of (1) Atlantic, (2) Mediterranean and (3) eastern European catchments, respectively.

| Region | No. of stations | Mean catchment area (km$^2$) | Mean outlet elevation (m a.s.l.) | Mean record length (years) |
|---|---|---|---|---|
| **1. Atlantic** | 855 | 1301.1 | 248.8 | 49.5 |
| **2. Mediterranean** | 382 | 2662.8 | 344.9 | 45.2 |
| **3. Eastern Europe** | 340 | 4667.4 | 104.8 | 49.6 |
| **Europe** | 2370 | 2472.3 | 286.0 | 48.8 |

**Table 1.** Regional summary statistics (number of stations, mean catchment area, mean outlet elevation, mean record length) of the 2370 selected stations in Europe. The regions 1-3 correspond to those shown in Fig. 1.


## 3 Results

### 3.1 Regional flood regime changes over central Europe

In this section we show a detailed example of the (local and) regional model estimates for the black rectangular 600x600 km window indicated in Fig. 1, located over central Europe and containing 601 hydrometric stations. The annual maximum discharge series of these stations are shown in Fig. 2 (thin lines and box-plots in panels a and b, respectively). In the same figure, the regional flood peak quantiles $q_2$ (panels a and b) and $q_{100}$ (panel b), estimated with Eq. 1, are shown (thick lines and shaded areas) as a function of time for five selected catchment areas (S=10, 100, 1000, 10 000 and 100 000 km$^2$, indicated by different colours) in panel a, and as a function of catchment area for 1985 (i.e. the median year of the analyses period) in panel b. In both panels, the 90% credible bounds (shaded areas) are shown together with the median value (thick lines) of the regional flood quantiles. Both $q_2$ and $q_{100}$ (not shown) increase in time and their trend is larger for smaller catchment areas. The uncertainties in the quantile estimates also vary with catchment area: for very small (e.g. 10 km$^2$) and very large (e.g. 100 000 km$^2$) catchments the credible bounds get larger, reflecting the scarcity of samples with these (extremely small and extremely large) size in the considered region (Fig. 1).

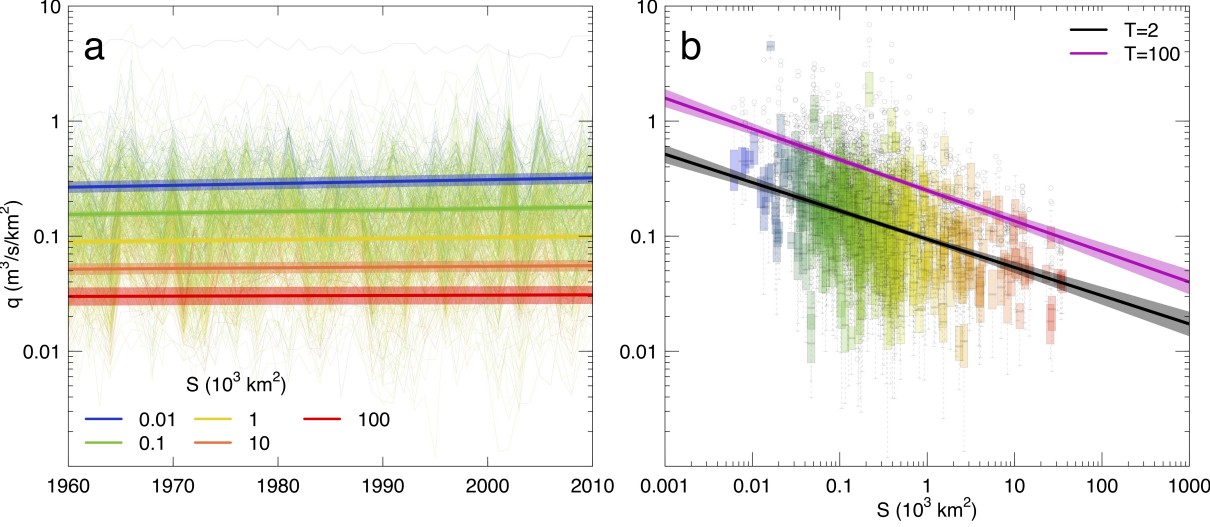

**Figure 2.** Fitting the regional model to flood data of the 601 hydrometric stations within the black rectangle shown in Fig. 1. In panel a, annual maximum specific discharge time series are shown with thin lines, with colours referring to catchment area. The thick lines and the shaded areas represent respectively the median and the 90% credible intervals of the estimated flood peak quantiles, corresponding to a return period of 2 years, for five hypothetical catchment areas (S=10, 100, 1000, 10 000 and 100 000 km$^2$, indicated by different colours). In panel b, the box-plots represent flood data as a function of catchment area. The thick lines and the shaded areas represent respectively the median and the 90% credible intervals of the estimated flood peak quantiles, with return period 2 and 100 years. The curves are shown for 1985, i.e. the median year of the period analysed.





The two panels of Fig. 3 show the relative change in time, in % per decade, of the regional flood quantile estimates $q_T$ (as defined in Eq. 3) as a function of catchment area and of the return period, respectively. The curves are shown for 1985, the median year of the analysed period. The trends in $q_T$ are mostly positive and their values tend to decrease with increasing

catchment area, approaching zero and moving towards negative values for higher return periods and for very large catchment areas (S=100 000 km$^2$). For small catchment areas (S<100 km$^2$) the trend tends to be bigger for floods with large return periods ($q_{100}$) than for small return periods ($q_2$). The opposite is observed for larger catchments. As in Fig. 2, we observe larger credible bounds of the quantile estimates for very small and very large catchment areas.

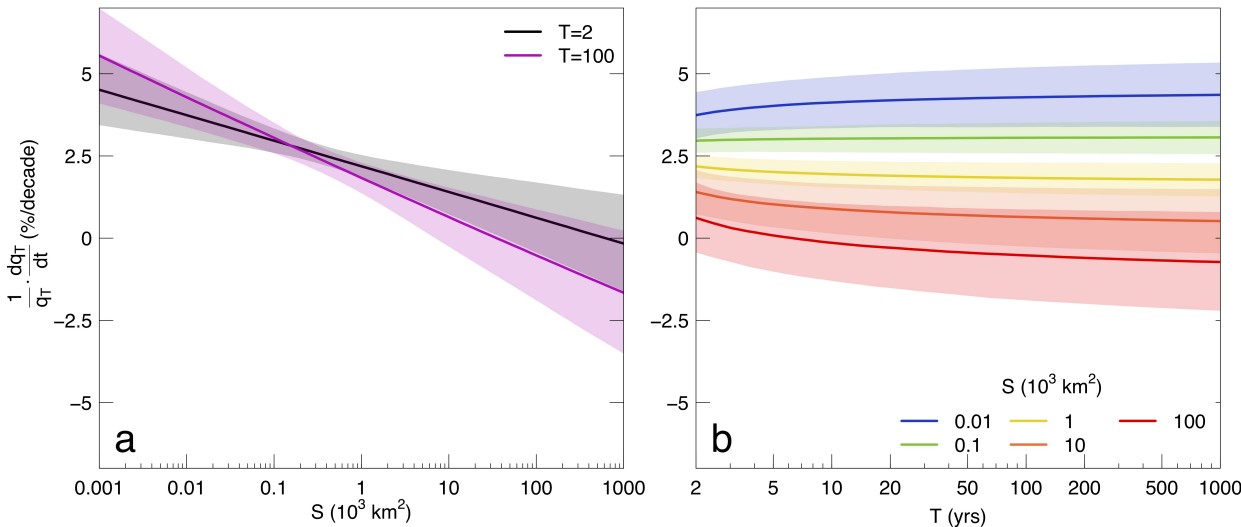

**Figure 3.** Estimates of the regional relative trend of $q_T$ in %/decade as a function of catchment area and return period. The thick lines and the shaded areas represent respectively the median and the 90% credible intervals of the estimated trends. Panel a shows the trend as a function of catchment area for selected values of the return period (T=2 and 100 years). Panel b shows it as a function of return period and for five hypothetical catchment area (S=10, 100, 1000, 10 000 and 100 000 km$^2$). The curves are shown for the median year of the period analysed (i.e. 1985). As in Fig. 2, the region considered is the black rectangle of Fig. 1.

Figure 4 summarizes the relative flood trends in the considered region for big vs. small floods (i.e. small return periods)

and for small to large catchment areas. It shows a scatter plot of the local relative trends in large ($q_{100}$) vs. small floods ($q_2$), as defined in Eq. 2, with the respective uncertainties (90% credible intervals) for 1985. On top of the local values, the regional relative trends, calculated with Eq. 3, are plotted. Again colours refer to catchment area for both the local and regional estimates. In Fig. 4 flood trends are generally positive in the considered region, with exception of big floods (T=100) in very large catchments (S=100 000 km$^2$). For both big and small events, the trend is generally larger in smaller catchments and it

diminishes with increasing catchment area, approaching zero, for small floods ($q_2$), and moving towards negative values for big floods ($q_{100}$, according to the credible intervals, we cannot determine if its trend for big catchments is different from zero).





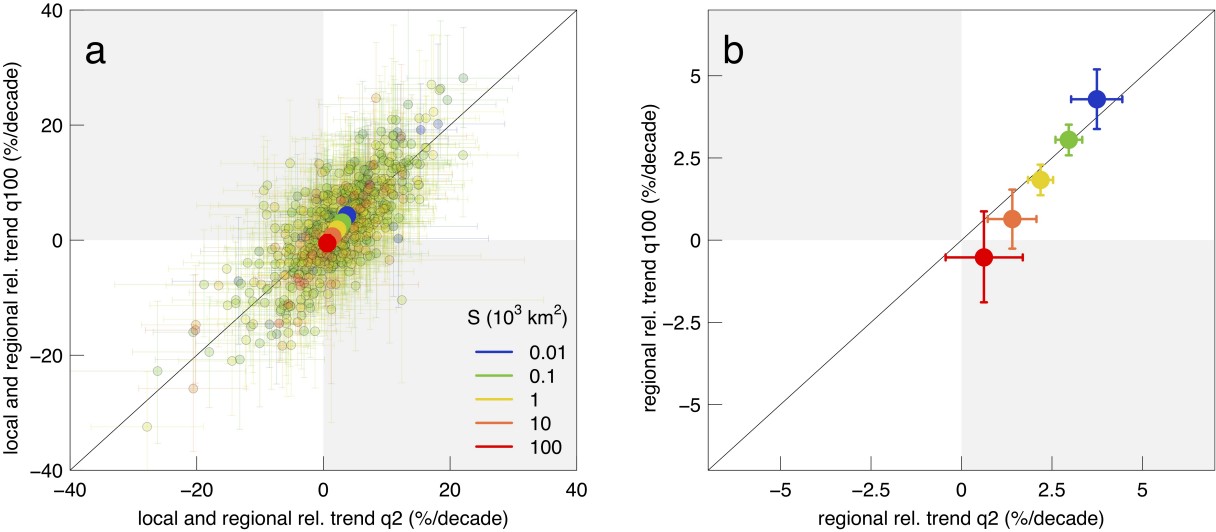

**Figure 4.** Local and regional relative trend in %/decade in large ($q_{100}$) vs. small floods ($q_2$). Panel a shows a scatter plot (light colour dots in the background) of the local relative trends in large ($q_{100}$) vs. small floods ($q_2$), with the 90% credible intervals (error bars). On top of them, the estimated median regional relative trends (solid points) are shown. Panel b shows the median regional relative trends (solid points) in large ($q_{100}$) vs. small floods ($q_2$), with the 90% credible intervals (error bars). Colours refer to catchment area in both panels and for both the local and regional estimates. The figure is obtained for 1985, i.e. the median year of the analyses period. As in Fig. 2, the region considered is the black rectangle of Fig. 1.

## 3.2 Regional flood regime changes across Europe

Figure 5 shows the results of the regional trend analysis with moving windows across Europe. It is obtained by fitting the regional model to overlapping 600x600 km windows and by plotting the estimated trend values in the respective central
200x200 km cell. Panels a and b show the percentage change of the median flood (i.e. T=2 years) and panels c and d of the 100-year flood. Panels a and c refer to small (i.e. 100 km$^2$) catchment area and panels b and d to big catchment area (i.e. 10 000 km$^2$). The white circles represent a measure of the uncertainty in the estimation of the regional relative trend, with their dimension being proportional to the width of the 90% credible intervals. The larger the circle, the larger the uncertainty associated with the value of flood trend provided in the map.

When analysing the panels of Fig. 5, some regional patterns of flood change appear: flood magnitudes increase in general in the British-Irish Isles, in north-western France and in central Europe, whereas they decrease in the Iberian peninsula, in the Balkans, in eastern Europe and in most of Scandinavian countries. The larger uncertainties associated with the regional trends are evident over eastern Europe, Turkey, Iceland and southern parts of the Mediterranean countries, where the density of the hydrometric stations in the flood database is low. In the British Isles and north-western France, the positive trends in
small catchments (up to 10-12% per decade, Fig. 5a and c) appear to be larger for bigger return periods (Fig. 5c), whereas for






**Figure 5.** Flood trends in Europe: small vs big floods. The panels show the median values of the regional relative trend of flood magnitude to time (i.e. the percentage change in %/decade). Positive trends in the magnitude of floods are shown in blue and negative trends in red. The circle size is proportional to the width of the 90% credible intervals. The results are shown for the median flood (i.e. T=2 years) in panel a and b, and for the 100-year flood in panel c and d. The flood trends refer to small (i.e. 100 km$^2$) in panel a and c and to large catchment area (i.e. 10 000 km$^2$) in panels b and d.





larger catchments the trends are smaller in absolute value (up to 5% per decade) and, in some cases, they tend to disappear or even to become negative. In central Europe, the magnitude of the positive trends (2.5-5% per decade) tends to decrease for large catchments and large return periods where, in most cases, the regional trends are between 0 and 2.5% per decade (Fig. 5b and d). Positive flood trends are also observed in northern Russia, especially in large catchments (Fig. 5b and d). These

positive trends are however accompanied by strong uncertainties in the case of small catchments (Fig. 5a and c). In the Iberian peninsula, south-western France, Italy and in the Balkans, negative trends appear and they are particularly consistent for the median floods (i.e. return period T=2 years), where the regional flood trends are mostly between -5 and -12% per decade (Fig. 5a and b). The trends in the magnitude of the big flood events (T=100 years) are less negative instead and some isolated positive trends do appear. The lower number of large catchments in this areas is in general reflected in larger uncertainties (Fig. 5b and

d). Over eastern Europe strong negative trends in flood peak magnitude are detected for big and small floods and small and large catchments. In eastern Europe, contrarily to the Mediterranean, the dataset contains mostly big catchments, hence the uncertainties are larger for small catchments (Fig. 5a and c). In Scandinavia the regional trends are, in general, not clearly positive nor negative with spatial patterns changing with return period and catchment area. However, in Finland negative trends are prevalent (mostly between -5 and -12% per decade) and they become less negative (0-5% per decade) for big catchments

and small return periods (Fig. 5b).

For further comparison, we estimate the differences between the regional relative trends in the panels of Fig. 5. In particular, Fig. 6a and 6b show the difference between the trend in $q_{100}$ and the trend in $q_2$, for big (i.e. $10\,000$ km$^2$) and small catchment area (i.e. $100$ km$^2$) respectively. Figure 6c and 6d show instead the difference between the trend in large and the trend in small catchments, for small (T=2 years) and big (T=100 years) return periods respectively. Positive differences are shown in blue and

negative in red. The circle size is proportional to the width of their 90% credible intervals.

In small catchments (Fig. 6a) positive differences between the trend in $q_{100}$ and in $q_2$ prevail in the British isles, the Iberian peninsula and southern France, the Balkans, eastern Europe and northern Russia. This indicates that, in the small catchments of these regions, the trend of the extreme flooding events is more positive (or less negative) compared to the median flood. Negative differences appear instead in central Europe, Baltic countries, southern Scandinavia and Turkey. The magnitude of

this difference varies in a narrow range (-2.5 to +2.5 % per decade) in most parts of Europe and it gets larger (up to -12 to +12 % per decade) in a number of regions in southern and eastern Europe

In the case of big catchments (Fig. 6b), negative differences between the trend in $q_{100}$ and in $q_2$ are more widespread across Europe, compared to the case of smaller catchments. In the British isles, southern France, north-western Italy, eastern Europe and northern Russia the difference becomes in fact negative. This suggests that, in the big catchments of these regions, the

trend of the extreme flooding events is less positive (or more negative) compared to the median flood. Positive values of this difference instead hold mostly in southern Europe and Russia. The magnitude of these differences, in the case of big catchments, varies in a wider range (generally from -5 to +5 % per decade) with larger differences in few regions in southern and eastern Europe.

The patterns appear more fragmented when analysing the differences between trends in catchments with big and small

catchment area (Fig. 6c and d) and their magnitude is generally larger (mostly from -12 to +12 % per decade). Negative





**Figure 6.** Differences between flood trends of big vs small floods (i.e. T=100 and 2 years, respectively) and in large vs small catchments (i.e. S=10 000 and 100 km², respectively). The panels show the differences (in % per decade) between the trends of Fig. 5. Positive differences are shown in blue and negative in red. The circle size is proportional to the width of the 90% credible intervals. The panels in the first row show the difference between the trend in $q_{100}$ and the trend in $q_2$ for small (a) and big catchment area (b). The panels in the second row show the difference between the trend in large and the trend in small catchments for small (c) and big (d) return periods.





differences between trends in large and trends in small catchments prevail in western and central Europe (with the exception of France), for both the median and the 100-year flood, and they extend towards eastern countries, in the case of the 100-year flood (Fig. 6d). This indicates that trends in large catchments are more negative (or less positive) than in small catchments. Positive differences appear instead in central and southern France, in the Balkans, Baltic countries and northern Russia, for

both T=2 and 100 years (Fig. 6c and d), and in Finland and eastern Europe, for T=100 years (Fig. 6d).

### 3.3  Regional flood regime changes in the Atlantic, Mediterranean and Eastern European regions

The regional trends shown in Sect. 3.2 highlight the presence of mostly positive trends in the Atlantic region and negative trends in the Mediterranean and over Eastern Europe. In this section we fit the regional model of Sect. 2.1 by pooling flood data over each of these three regions and we estimate the regional relative trends for five hypothetical catchment areas (S=10,

100, 1000, 10 000 and 100 000 $km^2$) and for two selected values of return period (T=2 and 100 years). The resulting trends are shown together with their 90% credible intervals in Fig. 7.

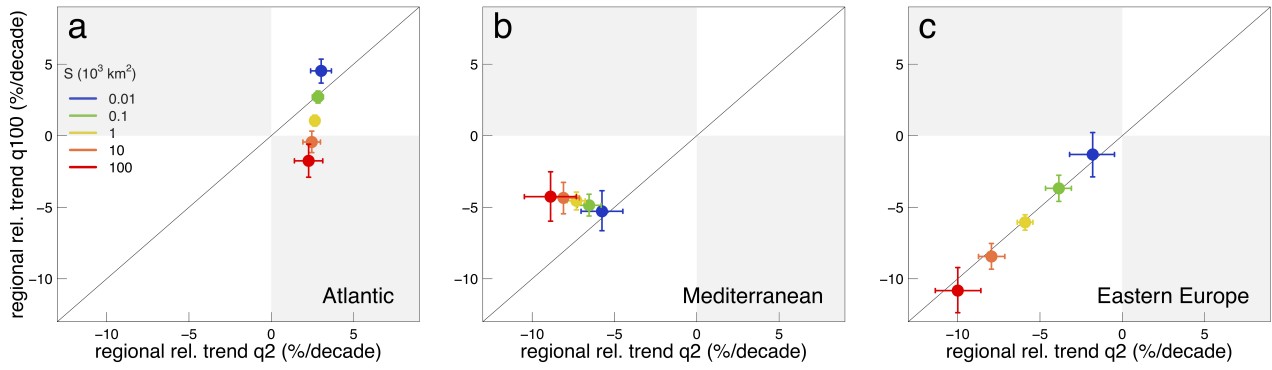

**Figure 7.** Regional relative trend in large ($q_{100}$) vs. small floods ($q_2$) in the Atlantic (a), Mediterranean (b) and eastern European (c) regions. The figure shows the median regional relative trends (solid points) together with their 90% credible intervals (error bars). Catchment area is shown with different colours. The figure is obtained for 1985, i.e. the median year of the period analysed.

In the Atlantic region (Fig. 7a) the trends in flood magnitudes are mainly positive, with the exception of very large catchments for the 100-year flood. The magnitude of the positive trend tends to decrease with increasing catchment area for the 2-year flood, while for the 100-year flood the positive trend decreases, it goes to zero for catchment size of about 10 000 $km^2$, and then

becomes negative and increases in absolute value for increasing catchment area. The trends are in general bigger for the 2-year flood compared to the 100-year flood, with exception of very small catchments (S=10 $km^2$). Overall there is large variability of the trend in $q_{100}$, which ranges from about -2.5 to 5 % per decade with catchment area, while the trend in $q_2$ is around 2 % per decade for all areas considered.

In the Mediterranean catchments the trends are negative in all the considered cases and larger in absolute value for the 2-year

flood. This means that the more frequent flood events tend to decrease more than the rare, more extreme events. In the smaller





catchments the regional relative trends in $q_2$ and $q_{100}$ are both about -5 % per decade. As catchment area increases, the trend in $q_2$ decreases from -5 to -10% per decade, while the trend in $q_{100}$ increases slightly from about -5.3 to -4.3 % per decade.

Over Eastern Europe the regional relative trends are all negative. The estimates lay close to the 1:1 line; this means that the trends are roughly the same for big and small events and that there is no variability with the return period. Catchment area
plays instead a more important role in determining flood trends. The magnitude of the negative trend appear, in fact, to be very sensitive to the catchment size and ranges from about -10.8 % per decade for bigger catchments, to 1.3 % per decade for smaller ones.

In all the regions analysed, it is also evident that the uncertainties in the trend estimates vary with catchment area: the credible bounds are in fact narrower for middle sized catchments that are represented by more hydrometric stations in the database.

**4    Discussion and conclusions**

In this study we assess and compare the changes occurred in small and big flood events and in small to large catchments across Europe, during the last five decades. Flood peaks are assumed to follow a regional Gumbel distribution, accounting for time dependency of two parameters alternative to the location and scale parameters: the 2-year flood $q_2$ and the 100-year flood growth factor $x'_{100}$. In flood frequency analysis, the Generalized Extreme Value distribution (GEV) is commonly used to
estimate flood quantiles (e.g. the 100-year flood). The suitability of the GEV distribution in the European context is discussed in detail in Salinas et al. (2014b, a). The estimate of the shape parameter of the GEV distribution is extremely sensitive to record length (Papalexiou and Koutsoyiannis, 2013), with strong bias and uncertainty for short records (Martins and Stedinger, 2000), and, when corrected for the effect of record length, it varies in a narrow range (Papalexiou and Koutsoyiannis, 2013). For these reasons, in regional frequency analyses the GEV shape parameter is commonly assumed to be identical for all sites
within a region (see e.g. Renard et al., 2006a; Lima et al., 2016). Here, we fix the shape parameter equal to 0 (i.e. we assume a Gumbel distribution) which leads to more robust relationships, without compromising the general validity of the study (i.e. the analysis can be repeated with a more complex GEV distribution if longer flood records are available). A Bayesian Monte Carlo Markov Chain (MCMC) approach is used for parameter estimation, allowing to directly obtain information about their associated uncertainties. Spatial cross-correlation between flood timeseries at different sites is not accounted for in this study
and may affect the estimation of sample uncertainty (see e.g. Stedinger, 1983; Castellarin et al., 2008; Sun et al., 2014). Because of this, the sample uncertainties estimated in this paper should be consider as a lower boundary. We expect that the effect of spatial correlation on the identified spatial patterns is negligible, since the cross-correlation length is about 50 km which is much shorter than the size of the spatial patterns.

We analyse 2370 flood records, selected from a newly-available pan-European flood database (Hall et al., 2015). We estimate
regional trends (and the related uncertainties) in the magnitude of floods of selected return periods (T=2 and 100 years) and for selected catchment areas (S=10 to 100 000 km$^2$), by fitting the proposed regional flood change model to flood data pooled within defined regions. Firstly, the trend patterns are investigated at the continental scale, by fitting the model to 600x600 km$^2$ overlapping windows, with a spatial moving window approach. Flood trends are then analysed in three regions (Atlantic,





Mediterranean and eastern European catchments, respectively), emerging from these change patterns. When fitting the model
to these regions, we allow for local spatial variations in the median but assume homogeneity with regards to the growth curves
of flood peaks, to changes in time and the dependency of the trends on catchment area and on the return period. The assumption
is that these regions are characterized by comparatively homogeneous climatic conditions and processes driving flood changes.
We have not assessed the statistical homogeneity of the regions in terms of the flood change model used here. One reason is
that formal procedures to assess the regional homogeneity, such as for example those used in regional flood frequency analysis
(e.g. Hosking and Wallis, 1993; Viglione et al., 2007), are not available at the moment. Also, while deviation from regional
homogeneity would probably invalidate estimates of local flood change statistics from the regional information (e.g., as in
the prediction in ungauged basins, see Blöschl et al., 2013b), we expect its effect on the average regional behavior to be less
relevant. This is because we have not observed significant differences in the regime changes when changing the size of the
moving windows (not shown here). As a limiting case, the results obtained using the three large climatic regions (Sect. 3.3) are
consistent with those obtained by the moving window analysis across Europe (Sect. 3.2).

The results of this study show that the trends in flood magnitude are generally positive in the Atlantic region, where floods
occur predominantly in winter (Mediero et al., 2015; Blöschl et al., 2017; Hall and Blöschl, 2018). The increasing winter
runoff in UK is typically explained by increasing winter precipitation and soil moisture (Wilby et al., 2008). Recent studies
show that extreme winter precipitation and flooding events over north-western Europe are positively correlated with the North
Atlantic Oscillation (NAO) and the East Atlantic (EA) pattern (Hannaford and Marsh, 2008; Steirou et al., 2019; Zanardo et al.,
2019; Brady et al., 2019). Furthermore the largest winter floods in Britain occur simultaneously with Atmospheric Rivers (AR)
(Lavers et al., 2011), which are expected to become more frequent in a warmer climate (Lavers and Villarini, 2013). When
comparing trends in flood events associated with different return periods, we observe two opposite behaviours depending on
catchment area. In small catchments (up to 100 km$^2$) the 100-year flood increases more than the median flood, while the
opposite is observed in medium and large catchments, where even some negative trends appear, especially over the continental
part of the Atlantic region. Furthermore, in medium and large catchments the magnitude of the trends is in general smaller
compared to the small catchments. This may be due to long-duration synoptic weather events, producing floods in medium
and large catchments, in contrast to small catchments in western Europe where the largest peaks are often caused by summer
convective events with high local intensities (Wilby et al., 2008), which are expected to increase in a warmer climate (IPCC,
2013).

In the Mediterranean catchments flood trends are negative due to decreasing precipitation and soil moisture, caused by
increasing evapotranspiration and temperature (Mediero et al., 2014; Blöschl et al., 2019). The big flood events (i.e. $T$=100
years) decrease less in time compared to more frequent events (i.e. $T$=2 years), leading to higher flood variability and steeper
flood frequency curves. The reason for this is likely (decreasing) soil moisture driving flood changes in southern Europe,
causing dryer catchments and consequent negative trends in flood magnitudes, that are particularly strong for small floods ($q_2$)
where the influence of soil moisture is stronger. The magnitude of big flood events is also decreasing (as an effect of decreasing
precipitation) but in this case soil moisture is less influential, resulting in less strong negative trends compared to $q_2$. In small
catchments, trends are less negative than in larger catchments for small return periods. For large return periods, the trends in





small and large catchments are similar. Additionally, in the smaller catchments we observe the same negative trend in $q_2$ and

$q_{100}$. Notice however that the small catchments analysed in the Mediterranean region have sizes of the order of 10 km$^2$, and are therefore larger than catchments where flash floods are the dominant flood type and infiltration excess runoff is the main generation mechanism (Amponsah et al., 2018). For these very small catchments ($< 10$ km$^2$), floods may become larger due to more frequent thunderstorms (Ban et al., 2015) and land management changes, e.g. deforestation and urbanisation (Rogger et al., 2017).

Over eastern Europe the trends in flood peak magnitude are strongly negative for both big and small floods, and large and small catchments. These negative flood trends have been linked with increasing spring air temperature, earlier snow-melt and reduced spring snow-cover extents (Estilow et al., 2015), producing increased infiltration and consequent earlier and decreasing spring floods (Madsen et al., 2014; Blöschl et al., 2017, 2019). The resulting trends in eastern Europe do not seem to depend on the return period (i.e. for a given catchment area, the trend in $q_2$ and the trend in $q_{100}$ are almost identical), whereas catchment

area plays a substantial role: the larger the catchment area, the more negative the trend. This results suggest that, in these region, snow-melt affects flood events of different magnitude in the same way and it represents a relevant processes for flood (trend) generation especially in large catchments. The explanation for that could be found in the characteristics of snow-melt flooding, which originates from large-scale gradual processes, i.e. snowfall and temperature changes, that may be more influential for large scale events, compared to smaller-scale catchments, where other local conditions may prevail.

The uncertainty associated with the regional trend estimates is here assessed through their 90 % credible bounds. The results show that the uncertainties in the trend estimates varies with catchment area: the credible bounds are in fact generally narrower for middle sized catchments, that are represented by more samples in the database, and they become wider for very small and very large values of catchment area. Spatial patterns in trend uncertainties are also observed. As expected, the uncertainty is lower in the regions where the density of stations is very high (i.e. central Europe and UK), while the estimated trend is very

uncertain over the data-scarce regions (i.e. southern and eastern Europe).

This study provides a continental-scale analysis of the changes in flood quantiles occurred over the last five decades across Europe. Our findings are relevant to flood risk managers to understand the amount of the potential over- or underestimation of the design flood at different spatial scales and in different European regions, when past flood changes are not taken into account. According to climate projections, the past flood regime changes found in this study, will further occur in next decades,

led by precipitation increase over Western Europe, decrease over the Mediterranean and temperature increase (see e.g. Alfieri et al., 2015; Kundzewicz et al., 2016; Thober et al., 2018). This has relevant implications since flood management has to adapt to these new realities.

*Data availability.* The flood discharge data used in this paper are available at https://github.com/tuwhydro/europe_floods. Data regarding catchment areas belong to different institutions listed in Extended Data Table 1, Blöschl et al. (2019).





*Author contributions.*  GB conceived the original idea, and all co-authors designed the overall study. AV developed the model. MB performed the analysis and prepared the paper. All co-authors contributed to the interpretation of the results and writing of the paper.

*Competing interests.*  The authors declare they have no conflict of interest.

*Acknowledgements.*  The authors would like to acknowledge funding from the European Union's Horizon 2020 Research and Innovation Programme under the Marie Skłodowska-Curie grant agreement No 676027, the FWF Vienna Doctoral Programme on Water Resource
Systems (W1219-N28) and the Austian Science Funds (FWF) "SPATE" project I 3174.



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
