# Peer review of "Flood trends in Europe: are changes in small and big floods different?"

_Hydrology and Earth System Sciences, 2019_

## Referee Comment (RC1) · Dominik Paprotny (Referee) · 5 Nov 2019

The manuscript "Flood trends in Europe: are changes in small and big floods different?" analysed changes in return periods of extreme river discharges between 1960 and 2010. The study is largely a follow-up to "Changing climate both increases and decreases European river floods" by Blöschl et al. (I will refer to it, for brevity, as the "Nature" paper). This doesn't compromise the novelty or importance of the submission, which is overall a well-written and important contribution. I have three major comments, and some very minor points.

Major comments:

1. The analysis in section 3.2 includes the uncertainty ranges of the trends, but their

ranges look in most cases proportional to the magnitude of the trend. I therefore find it not informative. It would be much more clear if instead of showing the uncertainty, to providing information whether the trends are statistically significant (at alpha of 0.1, or 0.05) by recolouring cells with insignificant trends grey. The text in section 3.2 could then be adjusted accordingly to the modified figures 5 & 6. The problem needs to be addressed as in the Nature paper as much as 72% of station trends were found to be insignificant. Also, in some areas the large uncertainty comes from the very limited number of stations. Though the stations are shown in Figure 1, a supplemental figure with the number of stations included in each 600 km box could be added, maybe even separately for large and small catchment sizes. This extra figure(s) is only a recommendation.

2. The analysis in section 3.3 includes three manually derived regions, which creates several problems. For one thing, no proper explanation for the choice is given. The Nature paper is cited as the source, but that paper also gives no real explanation apart for the attempt for homogenic regions (elliptical and overlapping for some reason). The other cited reference, Kotlarski et al. (2014) shows very different regional divisions (and not "not dissimilar" – btw. please avoid double negation). The regions omit, according to Table 1, one-third of stations in Europe, including most of the Danube catchment and northern Europe. Further, for some reason, the number of stations in each region is different than in the Nature paper, despite the ellipses being the same and the total number of stations as well. In summation, the authors should make a new derivation of regions, preferably based on actual geographical divisions of Europe (Fennoscandia, East European Plain, etc.), Koeppen's climate zones or drainage divides. Alternatively, cluster analysis could be used for this purpose. This would provide better connection between climate, topography and observed trends.

3. Not really a comment on paper, but an important question to the authors nonetheless. The authors provided an online dataset, and I noticed that it was updated recently in order to fix the errors in station coordinates. I wonder whether those errors affected

the paper's results and figures in any way, and whether they could account for the difference between the number of stations in Table 1 and the Nature paper. I suggest the authors check their data and code to ensure that there is no data-processing error present in their paper.

Minor comments:

Title: the study deals with floods understood as extreme river discharge, rather than floods as occurrence of losses. I know that's the hydrological vs natural hazards perspective issue, but even in HESS, the title could be more precise by mentioning "Flood discharge trends" rather than "Flood trends".

L4, L41, L128: the flood database is mentioned as "newly-available", but it has been compiled 4 years ago already. If it wasn't released publicly recently, the "newly-available" moniker should be removed.

L20-21: please correct this sentence, it's very ungrammatical.

L41-L44: when referring to the Nature paper, the names of regions from this submission are used instead of the Nature study. Especially the location of the "Atlantic" region, used throughout (including the abstract), is unclear until section 2.3.

Section 2.1: the similarity report noticed some overlaps in text with the author's other recent paper, which is not cited. Some comment in the section whether the presented methodology was used before or not would be beneficial.

L134-135: the authors repeat the explanation of station selection from the Nature paper, but given the methodological differences between the papers, I think the need for more even spatial distribution is much reduced here. Maybe some better explanation would do here.

L216: "British-Irish Isles" should be replaced with "British Isles", as this term encompasses Ireland.

L410: the reference to the Nature paper should be updated, as it is no longer "under review".

I am looking forward to the authors' revision of their paper.
* * *

---

## Author Comment (AC1) · 18 Nov 2019

**"Flood trends in Europe: are changes in small and big floods different?"**

by Miriam Bertola, Alberto Viglione, Julia Hall and Günter Blöschl

We reproduce in the following document all the comments of the Referees in *italic characters*, followed by our answers.

Referee #1: Dominik Paprotny

*The manuscript "Flood trends in Europe: are changes in small and big floods different?" analysed changes in return periods of extreme river discharges between 1960 and 2010. The study is largely a follow-up to "Changing climate both increases and decreases European river floods" by Blöschl et al. (I will refer to it, for brevity, as the "Nature" paper). This doesn't compromise the novelty or importance of the submission, which is overall a well-written and important contribution. I have three major comments, and some very minor points.*

> We want to thank the Referee Dominik Paprotny for the time he spent on our manuscript and for the useful and constructive comments. We have carefully considered and addressed all his comments in the following.

*Major comments:*

*1. The analysis in section 3.2 includes the uncertainty ranges of the trends, but their ranges look in most cases proportional to the magnitude of the trend. I therefore find it not informative. It would be much more clear if instead of showing the uncertainty, to providing information whether the trends are statistically significant (at alpha of 0.1, or 0.05) by recolouring cells with insignificant trends grey. The text in section 3.2 could then be adjusted accordingly to the modified figures 5 & 6. The problem needs to be addressed as in the Nature paper as much as 72% of station trends were found to be insignificant. Also, in some areas the large uncertainty comes from the very limited number of stations. Though the stations are shown in Figure 1, a supplemental figure with the number of stations included in each 600 km box could be added, maybe even separately for large and small catchment sizes. This extra figure(s) is only a recommendation.*

> We thank the Referee for raising this issue; we understand it deserves additional explanation and changes to the manuscript. In the Authors' opinion, it is more informative to show the uncertainty associated with the estimated regional flood trends (represented in figures 5 and 6 through the width of the 90% credible bounds), rather than discarding the statistically not significant trends. This is because, on the one hand, we are interested in showing the absence of the trend when it is associated with small uncertainties (i.e. cells where the estimated trend is close to zero and the credible bounds are narrow). In this case, the trend would result as statistically not significant from a trend test and the corresponding pixel would be shown in grey, as for those pixels where there is not enough information to reject the null hypothesis. In reality we have accurate information about the absence of the trend. On the other hand, when the estimated trend is statistically significant, but it is associated with very large uncertainties (e.g. in figure 5 in eastern Europe), we are interested in showing how much this estimate is uncertain (i.e. the width of the credible bounds). For these reasons we do not think that redrawing Figures 5 and 6 with grey pixels will improving them. However we will add to the text the number of pixels for which the 90% credible bounds do not include the value of zero trend, which is analogous to performing a trend test with alpha 0.1 (even though, strictly speaking, null-hypothesis significance testing is not a tool of Bayesian statistics). We will update section 3.2 accordingly.

> As the Referee correctly says, there is a tendency of having larger uncertainties in the regions where the trend magnitude (in absolute value) is larger. The Authors believe that this is due to extrapolation of the trends to catchment sizes not well represented in these regions, which results in large flood trend magnitudes (in

absolute value) which are indeed very uncertain. According to the Referee's suggestion, we will add one extra figure showing the number of stations in each cell, with a distinction of catchment size.

Furthermore, thanks to this comment we understand that figures 5 and 6, as they are, drive the attention of the reader to the stronger trends only (represented by the darker colours), without giving the right weight to the uncertainties (represented by the white circles). We will try to produce an alternative representation of figures 5 and 6 that better balances between the importance of the two type of information, i.e., the estimated flood trends and their uncertainties.

*2. The analysis in section 3.3 includes three manually derived regions, which creates several problems. For one thing, no proper explanation for the choice is given. The Nature paper is cited as the source, but that paper also gives no real explanation apart for the attempt for homogenic regions (elliptical and overlapping for some reason). The other cited reference, Kotlarski et al. (2014) shows very different regional divisions (and not "not dissimilar" – btw. please avoid double negation). The regions omit, according to Table 1, one-third of stations in Europe, including most of the Danube catchment and northern Europe. Further, for some reason, the number of stations in each region is different than in the Nature paper, despite the ellipses being the same and the total number of stations as well. In summation, the authors should make a new derivation of regions, preferably based on actual geographical divisions of Europe (Fennoscandia, East European Plain, etc.), Koeppen's climate zones or drainage divides. Alternatively, cluster analysis could be used for this purpose. This would provide better connection between climate, topography and observed trends.*

In the Nature paper, the 3 elliptical regions were identified by visual inspection of the flood trend patterns and by the selection of large homogeneous regions in terms of changes in the mean annual flood discharges. In this study (which, as the Referee correctly says, is a follow-up of the Nature paper), the same regions are selected because of consistency with this previous publication. Section 3.3 shows, in fact, changes in flood quantiles and the effect of catchment area for the 3 elliptical regions, that were found to be homogeneous in terms of changes in the mean annual flood discharges. We will clarify this by better explaining the reason for this choice in section 2.3. Moreover, we will correct the sentence with the double negation at line 158-160.

We thank the Referee for noticing the difference in terms of number of stations in each region in table 1. We will correct the table and reproduce figure 7 with the correct number of stations. We will also follow the Referee's suggestion to repeat the analysis of section 3.3 with pre-defined climate zones and we will add this analysis to the manuscript, if meaningful outcomes are found.

*3. Not really a comment on paper, but an important question to the authors nonetheless. The authors provided an online dataset, and I noticed that it was updated recently in order to fix the errors in station coordinates. I wonder whether those errors affected the paper's results and figures in any way, and whether they could account for the difference between the number of stations in Table 1 and the Nature paper. I suggest the authors check their data and code to ensure that there is no data-processing error present in their paper.*

We thank the Referee for spotting this and allowing us to check. The correction of the released flood data did not affect our analysis, as we used the original data. This correction was about an error occurred while producing the csv file for the public release.

**Minor comments:**

*Title: the study deals with floods understood as extreme river discharge, rather than floods as occurrence of losses. I know that's the hydrological vs natural hazards perspective issue, but even in HESS, the title could be more precise by mentioning "Flood discharge trends" rather than "Flood trends".*

We understand the Referee's point of view; we will clarify and emphasize in the abstract the fact that this study analyses river flood discharges. However, we would like to keep the original title.

*L4, L41, L128: the flood database is mentioned as "newly-available", but it has been compiled 4 years ago already. If it wasn't released publicly recently, the "newlyavailable" moniker should be removed.*

In the manuscript (L41 and L128) we cite an article about the compilation of the flood database (i.e. Hall et al., 2015) which was, at that time, ongoing. The flood database was instead publicly released in August 2019.

*L20-21: please correct this sentence, it's very ungrammatical.*

We will rephrase the sentence.

*L41-L44: when referring to the Nature paper, the names of regions from this submission are used instead of the Nature study. Especially the location of the "Atlantic" region, used throughout (including the abstract), is unclear until section 2.3.*

We will revise the manuscript and the abstract, in order to make sure that the locations of these three regions are clear to the reader from the beginning. We will additionally change the naming "Atlantic region" into "North-western Europe".

*Section 2.1: the similarity report noticed some overlaps in text with the author's other recent paper, which is not cited. Some comment in the section whether the presented methodology was used before or not would be beneficial.*

The main similarity with the Author's other recent publication is in the description of the tool used (i.e. rstan). Thank you for noticing it; we will rephrase the sentences at lines 119-122.

*L134-135: the authors repeat the explanation of station selection from the Nature paper, but given the methodological differences between the papers, I think the need for more even spatial distribution is much reduced here. Maybe some better explanation would do here.*

Since this work is a follow-up of the Nature paper and complementary analyses are presented, we should be consistent with this previous study by analysing the same flood data, in order to produce comparable results. We will better explain it in section 2.2.

*L216: "British-Irish Isles" should be replaced with "British Isles", as this term encompasses Ireland.*

We will correct it.

*L410: the reference to the Nature paper should be updated, as it is no longer "under review".*

Thank you for spotting it. The reference has been updated.

*I am looking forward to the authors' revision of their paper.*

We want to thank the Referee for his comments that will help to improve the quality of the manuscript.

**References:**

Hall, J. *et al.* (2015) 'A European flood database: Facilitating comprehensive flood research beyond administrative boundaries', *IAHS-AISH Proceedings and Reports*, 370, pp. 89–95. doi: 10.5194/piahs-370-89-2015.

---

## Referee Comment (RC2) · Duncan Faulkner (Referee) · 20 Nov 2019

The paper makes an interesting and valuable contribution to the large volume of literature on trends in flood magnitude. The pan-European focus is particularly valuable, as is the separation by flood rarity and catchment size.

Main comments The paper acknowledges that no allowance is made for spatial correlation of floods, and that this may affect the estimation of uncertainty. It claims that the regional model is more robust than the at-site trend analysis. This raises the question of the extent to which the apparent increase in robustness is due to the same information being repeated several times over, if trends at nearby gauges are reflecting essentially the same flood events. I recommend that the authors consider ways

of accounting for spatial correlation when quantifying uncertainty, such as a spatial nonparametric bootstrap or a likelihood correction (Sharkey and Winter, 2019). The authors quote a cross-correlation length of about 50km (section 4) which seems rather short in comparison with the spatial scale of some flood-producing weather systems.

The discussion (Section 4) makes various statements that go some way towards attributing trends. These vary from confident assertions (flood trends in the Mediterranean are negative due to . . .) to more informal or speculative comments using wording like "linked with", "suggest that", "could be found". I think many of us tend to use language like this when discussing trends, but in this context I would suggest the authors state more explicitly whether they are attempting a formal attribution of the trends or merely providing some hypotheses (or somewhere in between).

Minor comments The paper makes frequent use of the return period terminology. This is conceptually awkward in non-stationary conditions. I would suggest that the authors at least acknowledge this, and perhaps refer to some of the literature on alternative ways of expressing flood rarity.

The Gumbel parameters are modelled as varying with time according to a log-linear relationship. Perhaps the authors could comment on any alternative ways they considered of modelling trend, such as other mathematical forms of the relationship with time, or inclusion of physical covariates in an attempt to improve the identification of the time trends.

The meaning of gamma and S in the equations around lines 103-4 was not clear to me.

The assumption of homogeneity of the windows (section 2.3) seemed to me to need some justification.

I was impressed with the design of Figs 2 and 3, which pack in a great deal of information. I would suggest that the authors either remove or justify the extrapolation of the
model to catchment areas ten times smaller and ten times larger than those included in the dataset.

The description of Fig 5 mentions larger positive trends in NW France for big floods than for small floods. I could not see that effect from comparing the pairs of maps.

Reference Sharkey, P., & Winter, H. C. (2019). A Bayesian spatial hierarchical model for extreme precipitation in Great Britain. Environmetrics, 30(1), e2529.
* * *

---

## Referee Comment (RC3) · Duncan Faulkner (Referee) · 20 Nov 2019

To add to the debate - I support the original use by the authors of the term British-Irish Isles. Although this is less conventional, it is more politically accurate and less likely to cause offence in Ireland.

---

## Referee Comment (RC4) · Anonymous Referee #3 · 1 Dec 2019

The article is very nice, and contains a lot of nformation and results. One thing which is not clear from is the catchment size. FOr example, the Rijn has a catchment size of 180.000 km2, and contains also smaller catchments. How is this handled is this paper? Can smaller catchments be part of larger catchments? THis is important, because large catchments do show a negative trend. THis is not explained, and maybe there is no explanation, but has to be investigated in the future, this , however, can be stated more explicitly. THe following citation does NOT explain why the large catchments show different reults: "Furthermore, in medium and large catchments the magnitude of the trends is in general smaller compared to the small catchments. This may be due to long-duration synoptic weather events, producing floods in medium and large catchments, in contrast to small catchments in western Europe where the largest peaks

are often caused by summer convective events with high local intensities".

Does this suggests that "long-duration synoptic weather events" do show a negative trend?

---

## Author Comment (AC2) · 7 Jan 2020

**"Flood trends in Europe: are changes in small and big floods different?"**

by Miriam Bertola, Alberto Viglione, Julia Hall and Günter Blöschl

We reproduce and number in the following document all the comments of the Referee in *italic characters*, followed by our answers.

**Referee #2: Duncan Faulkner**

*The paper makes an interesting and valuable contribution to the large volume of literature on trends in flood magnitude. The pan-European focus is particularly valuable, as is the separation by flood rarity and catchment size.*

> We thank the Referee Duncan Faulkner for the time he spent on our manuscript and for the useful and constructive comments that will help to improve the quality of the manuscript. We have carefully considered and addressed all his comments in the following.

**Main comments**

*1. The paper acknowledges that no allowance is made for spatial correlation of floods, and that this may affect the estimation of uncertainty. It claims that the regional model is more robust than the at-site trend analysis. This raises the question of the extent to which the apparent increase in robustness is due to the same information being repeated several times over, if trends at nearby gauges are reflecting essentially the same flood events. I recommend that the authors consider ways of accounting for spatial correlation when quantifying uncertainty, such as a spatial nonparametric bootstrap or a likelihood correction (Sharkey and Winter, 2019). The authors quote a cross-correlation length of about 50km (section 4) which seems rather short in comparison with the spatial scale of some flood-producing weather systems.*

> The referee is right, spatial cross-correlation between flood timeseries at different sites is not accounted for in this study and it may affect the estimation of sample uncertainty. In particular, if the flood timeseries at different sites are strongly correlated, we expect the uncertainties to be larger than the uncertainties estimated in this paper. Therefore, at lines 299-301 we state that the estimated credible bounds should be intended as lower limits. However, we do not expect the trend estimates (i.e., in this case, the posterior median) to change, when cross-correlation is taken into account (Stedinger, 1983; Hosking and Wallis 1988 and 1997).

> We thank the Referee for suggesting possible ways to account for spatial correlation. In particular, the likelihood correction approach in Sharkey and Winter (2019) seems to fit this case, as the parameter estimates derived from the corrected likelihood are unchanged and the independence likelihood is scaled down, resulting in inflated asymptotic variance of the posterior. Based on the results of Sharkey and Winter (2019), we expect the credible bounds to be up to 20% wider with the adjusted likelihood. We will apply the likelihood correction approach for the example region of section 3.1, using the likelihood correction factor as estimated in Sharkey and Winter (2019), in order to quantify the magnitude of the increase in the width of the credible bounds when spatial cross-correlation is taken into account. Based on the result, we will also introduce additional text in the discussion section 4.

> The cross-correlation length of about 50 km has been calculated from the flood timeseries using distances between the catchment outlets. We will state more precisely how this has been calculated in the revised manuscript.

*2. The discussion (Section 4) makes various statements that go some way towards attributing trends. These vary from confident assertions (flood trends in the Mediterranean are negative due to ...) to more informal or speculative comments*

*using wording like "linked with", "suggest that", "could be found". I think many of us tend to use language like this when discussing trends, but in this context I would suggest the authors state more explicitly whether they are attempting a formal attribution of the trends or merely providing some hypotheses (or somewhere in between).*

Thank you for pointing this out; we understand that we need to clarify the nature of our statements in section 4. This work does not aim at formally attributing the observed flood trends to drivers and the statements in the discussion section, about the potential causes of the observed flood trends, are intended to be hypotheses or interpretation of the results, based on the literature and on the Authors' understanding of these processes. We will clarify the nature of our statements in section 4 and we will mitigate the statements that sound like confident assertations.

**Minor comments**

*The paper makes frequent use of the return period terminology. This is conceptually awkward in non-stationary conditions. I would suggest that the authors at least acknowledge this, and perhaps refer to some of the literature on alternative ways of expressing flood rarity.*

We understand that the use of the return period terminology in a non-stationary context may sound ambiguous to some readers. In the manuscript we refer to the return period (rather than to the annual exceedance probability) because, in the engineering practice, it is widely understood what a 100-year flood is. Examples of return period terminology used in a similar non-stationary context in the literature are Renard et al. (2006), Machado et al. (2015), Šraj et al. (2016). For these reasons we prefer to maintain the return period terminology in the manuscript. However, we will clarify the terminology used in the method section 2.1.

*The Gumbel parameters are modelled as varying with time according to a log-linear relationship. Perhaps the authors could comment on any alternative ways they considered of modelling trend, such as other mathematical forms of the relationship with time, or inclusion of physical covariates in an attempt to improve the identification of the time trends.*

Thank you for pointing this out. We will introduce additional text about alternative ways of modelling trends in section 4. The use of physical covariates in order to attribute the trends in flood quantiles is actually planned in the next phase of our research.

*The meaning of gamma and S in the equations around lines 103-4 was not clear to me.*

Thank you for spotting this lack of clarity in these lines. The gammas are parameters of the model that control the scaling with catchment area and S is catchment area. We will specify it in lines 103-107.

*The assumption of homogeneity of the windows (section 2.3) seemed to me to need some justification.*

Thank you for rising this point; we understand that we need to clarify it in section 2.3. As described in the manuscript in section 4 (lines 314-321), we have not formally tested this assumption (i.e. the statistical homogeneity of the 600x600 km regions in terms of the flood change model used here), because formal procedures to assess the regional homogeneity, such as for example those used in regional flood frequency analysis (e.g., Hosking and Wallis, 1993; Viglione et al., 2007), are not available at the moment. Furthermore, while deviation from regional homogeneity would probably invalidate estimates of local flood change statistics from the regional information (e.g., as in the prediction in ungauged basins, see Blöschl et al., 2013), we expect its effect on the average regional behavior to be less relevant. However, this assumption seems to have a small influence on the results, as we have not observed significant differences in the regime changes when changing the size of the moving windows (this was done in preliminary tests and is not shown in the results). We will introduce additional text in section 2.3 to clarify this point.

*I was impressed with the design of Figs 2 and 3, which pack in a great deal of information. I would suggest that the authors either remove or justify the extrapolation of the model to catchment areas ten times smaller and ten times larger than those included in the dataset.*

We agree with the Referee's suggestion of removing the extrapolation of the model to very small and very large catchment areas. We will modify Figures 2 and 3 accordingly.

*The description of Fig 5 mentions larger positive trends in NW France for big floods than for small floods. I could not see that effect from comparing the pairs of maps.*

Thank you for noticing it; we will correct the description of Figure 5.

**Reference**

Blöschl, G., Sivapalan, M., Wagener, T., Viglione, A., and Savenije, H. (2013) 'Runoff Prediction in Ungauged Basins: Synthesis across Processes, Places and Scales', Cambridge University Press. doi: 10.1017/CBO9781139235761.

Hosking JRM, Wallis JR. The effect of intersite dependence on regional flood frequency analysis. Water Resources Research 1988; 24:588-600.

Hosking, J. R. M. and Wallis, J. R.: Some statistics useful in regional frequency analysis, Water Resources Research, 29, 271–281, https://doi.org/10.1029/92WR01980, 1993.

Hosking JRM, Wallis JR. Regional Frequency Analysis: an approach based on L- Moments. Cambridge, UK: Cambridge University Press, 1997

Machado, M. J. *et al.* (2015) 'Flood frequency analysis of historical flood data under stationary and non-stationary modelling', *Hydrology and Earth System Sciences*, 19(6), pp. 2561–2576. doi: 10.5194/hess-19-2561-2015.

Renard, B., Lang, M. and Bois, P. (2006) 'Statistical analysis of extreme events in a non-stationary context via a Bayesian framework: case study with peak-over-threshold data', Stochastic Environmental Research and Risk Assessment, 21(2), pp. 97–112. doi: 10.1007/s00477-006-0047-4.

Sharkey, P., & Winter, H. C. (2019). A Bayesian spatial hierarchical model for extreme precipitation in Great Britain. Environmetrics, 30(1), e2529.

Stedinger JR. Estimating a regional flood frequency distribution. Water Resources Research 1983; 19:503-510.

Šraj, M. *et al.* (2016) 'The influence of non-stationarity in extreme hydrological events on flood frequency estimation', *Journal of Hydrology and Hydromechanics*, 64(4), pp. 426–437. doi: 10.1515/johh-2016-0032.

Viglione, A., Laio, F., and Claps, P.: A comparison of homogeneity tests for regional frequency analysis, Water Resources Research, 43, 545 1–10, https://doi.org/10.1029/2006WR005095, 2007.

---

## Author Comment (AC4) · 7 Jan 2020

Thank you for your comment, we will follow your suggestion and maintain the original terminology.

———————————————————

---

## Author Comment (AC5) · 7 Jan 2020

**"Flood trends in Europe: are changes in small and big floods different?"**

by Miriam Bertola, Alberto Viglione, Julia Hall and Günter Blöschl

We reproduce in the following document all the comments of the Referee in *italic characters*, followed by our answer.

**Anonymous Referee #3:**

*The article is very nice, and contains a lot of information and results. One thing which is not clear from is the catchment size. For example, the Rijn has a catchment size of 180.000 km2, and contains also smaller catchments. How is this handled in this paper? Can smaller catchments be part of larger catchments? This is important, because large catchments do show a negative trend. This is not explained, and maybe there is no explanation, but has to be investigated in the future, this , however, can be stated more explicitly. The following citation does NOT explain why the large catchment show different results: "Furthermore, in medium and large catchments the magnitude of the trends is in general smaller compared to the small catchments. This may be due to long-duration synoptic weather events, producing floods in medium and large catchments, in contrast to small catchments in western Europe where the largest peaks are often caused by summer convective events with high local intensities". Does this suggests that "long-duration synoptic weather events" do show a negative trend?*

We thank the Anonymous Referee #3 for the time she/he has spent on our manuscript and for the useful and constructive comments that will help to improve the quality of the manuscript. We have carefully considered and addressed her/his comments in the following.

In this study, we analyse flood data from 2370 hydrometric stations in Europe, each one corresponding to a specific catchment size/area. Consequently, multiple smaller catchments can be part of a larger catchment. The Gumbel regional model is fitted to flood records that are pooled over regions. In each region, the regional trend corresponding to a large (hypothetical) catchment area is an output of the model and is determined by the flood records, within the region, that correspond to large catchment areas. We will clarify it in section 2.2. The trend in large catchments is shown in Figure 5 and its sign and estimated magnitude varies according to the location and the flood quantile considered.

In this work we do not aim at investigating the causes of the observed flood trends (this is actually planned for the next phase of this research). However, in the discussion section, we make hypotheses on the possible drivers of the resulting flood trends, based on the literature and on our understanding of these processes. The sentences pointed out by the Referee (lines 331-334) refer to the Atlantic region, where large catchments exhibit, in general, smaller trends (positive for the 2-year quantile and negative for the 100-year quantile) compared to the smaller catchments. We did not intend to attribute this difference to specific drivers, nor to suggest negative trends in long-duration synoptic weather events. These aspects will be investigated in future work. In these lines we hypothesize that different type of weather events could affect in different ways the flood trends in catchments of different size. We will change these sentences (lines 331-334) accordingly.

---

## Author Response (AR1)

Response to the comments to the manuscript **hess-2019-523**

**"Flood trends in Europe: are changes in small and big floods different?"**

by Miriam Bertola, Alberto Viglione, David Lun, Julia Hall and Günter Blöschl

We wish to thank the Editor and the three Reviewers for the time they spent on our manuscript and for the positive and constructive comments. For the sake of convenience, we reproduce and number in the following document all the comments of the Reviewers in *italic characters*, followed by our answers. Together with the revised manuscript in PDF we also send the Marked Manuscript in PDF in which all the changes in the text are tracked (deleted in  characters, while new text is in blue characters). Numbers in brackets (highlighted in yellow) indicate the line numbers in the Marked Manuscript.
* * *
**Editor**

*Comments to the Author:*
*Dear Authors,*
*Three reviewers evaluated your manuscript and gave useful feedback to improve your manuscript. The overall evaluation is positive, although some changes/improvements/clarifications are needed. Please adjust the manuscript according to the reviewers suggestions. The manuscript will be sent out for another round of review.*
*Sincerely,*
*Albrecht Weerts*

> We thank the Editor Albrecht Weerts for this chance to improve our manuscript; we have carefully considered and addressed all the comments provided by the Reviewers as detailed in the following pages.
* * *
**Referee #1: Dominik Paprotny**

*The manuscript "Flood trends in Europe: are changes in small and big floods different?" analysed changes in return periods of extreme river discharges between 1960 and 2010. The study is largely a follow-up to "Changing climate both increases and decreases European river floods" by Blöschl et al. (I will refer to it, for brevity, as the "Nature" paper). This doesn't compromise the novelty or importance of the submission, which is overall a well-written and important contribution. I have three major comments, and some very minor points.*

> We want to thank the Referee Dominik Paprotny for the time he spent on our manuscript and for the useful and constructive comments. We have carefully considered and addressed all his comments in the following.

**Major comments:**

*1. The analysis in section 3.2 includes the uncertainty ranges of the trends, but their ranges look in most cases proportional to the magnitude of the trend. I therefore find it not informative. It would be much more clear if instead of showing the uncertainty, to providing information whether the trends are statistically significant (at alpha of 0.1, or 0.05) by recolouring cells with insignificant trends grey. The text in section 3.2 could then be adjusted accordingly to the modified figures 5 & 6. The problem needs to be addressed as in the Nature paper as much as 72% of station trends were*

*found to be insignificant. Also, in some areas the large uncertainty comes from the very limited number of stations. Though the stations are shown in Figure 1, a supplemental figure with the number of stations included in each 600 km box could be added, maybe even separately for large and small catchment sizes. This extra figure(s) is only a recommendation.*

We thank the Referee for raising this issue; we understand it deserves additional explanation and changes to the manuscript. In the Authors' opinion, it is more informative to show the uncertainty associated with the estimated regional flood trends (represented in figures 5 and 6 through the width of the 90% credible bounds), rather than discarding the statistically not significant trends. This is because, on the one hand, we are interested in showing the absence of the trend when it is associated with small uncertainties (i.e. cells where the estimated trend is close to zero and the credible bounds are narrow). In this case, the trend would result as statistically not significant from a trend test and the corresponding pixel would be shown in grey, as for those pixels where there is not enough information to reject the null hypothesis. In reality we have accurate information about the absence of the trend. On the other hand, when the estimated trend is statistically significant, but it is associated with very large uncertainties (e.g. in figure 5 in eastern Europe), we are interested in showing how much this estimate is uncertain (i.e. the width of the credible bounds). For these reasons we do not think that redrawing Figures 5 and 6 with grey pixels will improve them. However we have added to the text the number of pixels for which the 90% credible bounds do not include the value of zero trend, which is analogous to performing a trend test with alpha 0.1 (even though, strictly speaking, null-hypothesis significance testing is not a tool of Bayesian statistics). The additional sentences (lines 288-292) are: "Overall, in more than half of the cases the 90% credible bounds do not include 0 (i.e. 68.9%, 59.2%, 58.5% and 50.2% respectively in panel a, b, c and d). Positive (negative) trends occur in 26.3 to 34.95% (65 to 76%) of the cases and their credible bounds do not include zero in 4.9 to 20.8% (39.5 to 48.1%) of the total cells. These percentages depend on the assumptions made, such as regional homogeneity and no spatial cross-correlation, and may, therefore, be overestimated".

As the Referee correctly says, there is a tendency of having larger uncertainties in the regions where the trend magnitude (in absolute value) is larger. The Authors believe that this is due to extrapolation of the trends to catchment sizes not well represented in these regions, which results in large flood trend magnitudes (in absolute value) which are indeed very uncertain. According to the Referee's suggestion, we have one extra figure to the Appendix (Fig. A1) showing the number of stations in each 600x600 km cell, with a distinction of catchment size. Furthermore, thanks to this comment we understood that figures 5 and 6 drove the attention of the reader to the stronger trends only (represented by the darker colours), without giving the right weight to the uncertainties (represented by the white circles). We have therefore replaced figure 5 and 6 with an alternative representation that better balances between the importance of the two types of information, i.e., the estimated flood trends and their uncertainties.

*2. The analysis in section 3.3 includes three manually derived regions, which creates several problems. For one thing, no proper explanation for the choice is given. The Nature paper is cited as the source, but that paper also gives no real explanation apart for the attempt for homogenic regions (elliptical and overlapping for some reason). The other cited reference, Kotlarski et al. (2014) shows very different regional divisions (and not "not dissimilar" – btw. please avoid double negation). The regions omit, according to Table 1, one-third of stations in Europe, including most of the Danube catchment and northern Europe. Further, for some reason, the number of stations in each region is different than in the Nature paper, despite the ellipses being the same and the total number of stations as well. In summation, the authors should make a new derivation of regions, preferably based on actual geographical divisions of Europe (Fennoscandia, East European Plain, etc.), Koeppen's climate zones or drainage divides. Alternatively, cluster analysis could be used for this purpose. This would provide better connection between climate, topography and observed trends.*

In the Nature paper, the 3 elliptical regions were identified by visual inspection of the flood trend and flood seasonality patterns and by the selection of large homogeneous regions in terms of changes in the mean annual flood discharges. In this study (which, as the Referee correctly says, is a follow-up of the Nature paper), the same regions are analysed in terms of changes in flood quantiles, to allow a more detailed assessment of existing research and to allow for ready comparability of the results. Furthermore, the location and extension

of these three macro-regions are comparable with some of the climate zones of the Köppen-Geiger classification, suggested by the Referee. Northwestern Europe (region 1), in fact, roughly corresponds to the temperate oceanic climate zone, in southern Europe (region 2) hot and warm summer Mediterranean climate zones prevail, and eastern Europe (region 3) is dominated by the warm summer humid continental climate zone. For these reasons we decided to maintain the original regions, however, we have clarified this choice and corrected the sentence with double negation in lines 192-202: "Figure 1 shows three macro-regions (numbers 1-3) located over northwestern Europe, southern Europe and eastern Europe, respectively. These regions were identified in Blöschl et al. (2019) by visual inspection of the flood trend and flood seasonality patterns and represent large homogeneous regions in terms of changes in the mean annual flood discharges. According to Köppen-Geiger climate classification (Köppen, 1884), northwestern Europe (region 1) corresponds approximately to the temperate oceanic climate zone, in southern Europe (regions 2) the hot and warm summer Mediterranean climate zones prevail, and eastern Europe (region3) is dominated by warm summer humid continental climate. Table 1 shows some related regional summary statistics. In this study the same regions are analysed in terms of changes in flood quantiles, to allow a more detailed assessment of existing research and to allow for ready comparability of the results." We thank the Referee for noticing the difference in terms of number of stations in each region in table 1. We have corrected the table and reproduced figure 7 with the correct number of stations.

*3. Not really a comment on paper, but an important question to the authors nonetheless. The authors provided an online dataset, and I noticed that it was updated recently in order to fix the errors in station coordinates. I wonder whether those errors affected the paper's results and figures in any way, and whether they could account for the difference between the number of stations in Table 1 and the Nature paper. I suggest the authors check their data and code to ensure that there is no data-processing error present in their paper.*

We thank the Referee for spotting this and allowing us to check. The correction of the released flood data did not affect our analysis, as we used the original data. This correction was about an error occurred while producing the csv file for the public release.

***Minor comments:***

*Title: the study deals with floods understood as extreme river discharge, rather than floods as occurrence of losses. I know that's the hydrological vs natural hazards perspective issue, but even in HESS, the title could be more precise by mentioning "Flood discharge trends" rather than "Flood trends".*

We understand the Referee's point of view; we have clarified and emphasized in the abstract (lines 3-4) the fact that this study analyses river flood discharges, by changing the sentence from: "The aim of this study is to assess whether trends also occurred for larger return periods accounting for the effect of catchment scale. We analyze 2370 flood records […]" to "The aim of this study is to assess whether trends in flood discharges also occurred for larger return periods accounting for the effect of catchment scale. We analyze 2370 flood discharge records […]". However, we prefer to keep the original title.

*L4, L41, L128: the flood database is mentioned as "newly-available", but it has been compiled 4 years ago already. If it wasn't released publicly recently, the "newlyavailable" moniker should be removed.*

In the manuscript (L41 and L128) we cite an article about the compilation of the flood database (i.e. Hall et al., 2015) which was, at that time, ongoing. The flood database was instead publicly released in August 2019.

*L20-21: please correct this sentence, it's very ungrammatical.*

We have rephrased the sentence (lines 20-24) to: "Increasing flood hazard in Europe has become a major concern as a consequence of severe flood events experienced in the last decades, as, for instance, the extreme floods occurred in central Europe in 2002 (e.g. Ulbrich et al., 2003) and 2013 (e.g. Blöschl et al., 2013a), and the winter floods in northwest England in 2009 (e.g. Miller et al., 2013) and 2015-16 (e.g. Barker et al., 2016)."

*L41-L44: when referring to the Nature paper, the names of regions from this submission are used instead of the Nature study. Especially the location of the "Atlantic" region, used throughout (including the abstract), is unclear until section 2.3.*

> We have re-named the "Atlantic" and "Mediterranean" regions into "Northwestern Europe" and "Southern Europe" throughout the manuscript and the abstract.

*Section 2.1: the similarity report noticed some overlaps in text with the author's other recent paper, which is not cited. Some comment in the section whether the presented methodology was used before or not would be beneficial.*

> The main similarity with the Author's other recent publication is in the description of the tool used (i.e. rstan). Thank you for noticing it; we have rephrased the sentences at lines 134-139.

*L134-135: the authors repeat the explanation of station selection from the Nature paper, but given the methodological differences between the papers, I think the need for more even spatial distribution is much reduced here. Maybe some better explanation would do here.*

> Since this work is a follow-up of the Nature paper and complementary analyses are presented, we should be consistent with this previous study by analysing the same flood data, in order to produce comparable results. We have better explained it in lines 161-162: "For comparability with Blöschl et al. (2019), only the stations satisfying the following selection criteria, based on record length and even spatial distribution, are considered for the estimation of the regional trends".

*L216: "British-Irish Isles" should be replaced with "British Isles", as this term encompasses Ireland.*

> According to the interactive comment of the Referee #2, we have maintained the original inclusive term "British-Irish Isles".

*L410: the reference to the Nature paper should be updated, as it is no longer "under review".*

> The reference has been updated.

*I am looking forward to the authors' revision of their paper.*

> We want to thank the Referee for his comments that helped to improve the quality of the manuscript.
* * *
**Referee #2: Duncan Faulkner**

*The paper makes an interesting and valuable contribution to the large volume of literature on trends in flood magnitude. The pan-European focus is particularly valuable, as is the separation by flood rarity and catchment size.*

> We thank the Referee Duncan Faulkner for the time he spent on our manuscript and for the useful and constructive comments that helped to improve the quality of the manuscript. We have carefully considered and addressed all his comments in the following.

**Main comments**

*1. The paper acknowledges that no allowance is made for spatial correlation of floods, and that this may affect the estimation of uncertainty. It claims that the regional model is more robust than the at-site trend analysis. This raises the question of the extent to which the apparent increase in robustness is due to the same information being repeated several times over, if trends at nearby gauges are reflecting essentially the same flood events. I recommend that the authors consider ways of accounting for spatial correlation when quantifying uncertainty, such as a spatial nonparametric bootstrap or a likelihood correction (Sharkey and Winter, 2019). The authors quote a cross-correlation length of about 50km (section 4) which seems rather short in comparison with the spatial scale of some flood-producing weather systems.*

> The referee is right, spatial cross-correlation between flood timeseries at different sites is not accounted for in this study and it may affect the estimation of sample uncertainty. In particular, if the flood timeseries at

different sites are strongly correlated, we expect the uncertainties to be larger than the uncertainties estimated in this paper. Therefore, we state that the estimated credible bounds should be intended as lower limits (lines 386-389). However, we do not expect the trend estimates (i.e., in this case, the posterior median) to change, when cross-correlation is taken into account (Stedinger, 1983; Hosking and Wallis 1988 and 1997).

We thank the Referee for suggesting possible ways to account for spatial dependence in flood data. We have applied the magnitude adjustment to the likelihood (described in Sharkey and Winter, 2019) to the example region of section 3.1, in order to quantify, for this specific case, the increase in the width of the credible bounds when spatial cross-correlation is taken into account. Figures 3 and 4 have been updated accordingly. Additional text has been added to the manuscript:

- In section 2.1 (lines 144-154): "Spatial cross-correlation between flood time series at different sites is not accounted for in this model (i.e. it assumes independence of flood time series in space), however, it is possible to quantify its effects in first approximation in a Bayesian framework through an approach based on a magnitude adjustment to the likelihood (Ribatet et al., 2012). This approach consists in scaling the likelihood with a proper constant exponent to be estimated between 0 and 1, that results in inflating the posterior variance of the parameters and consequent increase of the width of parameter uncertainty intervals, reflecting the overall effect of spatial dependence in the data. In the case of spatial independence the magnitude adjustment factor is 1, whereas values of the magnitude adjustment factor close to 0 indicate strong inter-site correlation of floods and substantially larger sample uncertainties resulting from the adjusted model, compared to the model where spatial cross-correlation is not accounted for. For further details about the method and its application to hydrological data, see Smith (1990), Ribatet et al. (2012) and Sharkey and Winter (2019). We apply this method to an example region in central Europe, in order to quantify the magnitude of the uncertainty underestimation associated with the model assumption of spatial independence in flood data."
- In section 2.3 (lines 211-213): "In this example region we also investigate the effect of spatial dependence in flood data on the width of the estimated credible bounds, with an approach based on the magnitude adjustment to the likelihood".
- in section 3.1 (lines 243-246): "In this case, the overall effect of spatial cross-correlation between flood time series at different sites is investigated through the magnitude adjustment to the likelihood. The credible bounds of the regional trends obtained with the likelihood adjustment (dashed lines in Fig. 3) are 17.6 to 23.8% larger compared to the case where spatial cross-correlation is not accounted for (estimated magnitude adjustment factor 0.669)" and (lines 257-258) "The credible bounds obtained with the likelihood adjustment (dashed lines) result slightly wider (about 20%) compared to the case where spatial cross-correlation is not accounted for."
- in section 4 (lines 370-376): "A possible way of taking into account spatial cross-correlation between sites is a magnitude adjustment to the likelihood, that reflects the overall effect of spatial dependence and results in increased width of uncertainty intervals of the estimated quantiles (see Ribatet et al., 2012). The application of this approach to the specific example region in central Europe shows that the 90% credible bounds of the regional trends in q2 and q100 result, on average, 20% wider compared to the case where the likelihood is not adjusted. However, further research is needed to properly characterize the effect of spatial dependence between flood peaks in regional frequency analyses."

At lines 369-370 we have stated how the cross-correlation length of about 50 km has been calculated: "[…] calculated from flood timeseries and distances between catchment outlets using a nonlinear regression model proposed by Tasker and Stedinger, 1989 […]".

*2. The discussion (Section 4) makes various statements that go some way towards attributing trends. These vary from confident assertions (flood trends in the Mediterranean are negative due to …) to more informal or speculative comments*

*using wording like "linked with", "suggest that", "could be found". I think many of us tend to use language like this when discussing trends, but in this context I would suggest the authors state more explicitly whether they are attempting a formal attribution of the trends or merely providing some hypotheses (or somewhere in between).*

Thank you for pointing this out. This work does not aim at formally attributing the observed flood trends to drivers and the statements in the discussion section, about the potential causes of the observed flood trends, are intended to be hypotheses or interpretation of the results, based on the literature and on the Authors' understanding of these processes. We have mitigated the statements about the trend attribution that sound like confident assertations throughout the discussion section 4. Additionally, in order to avoid misunderstandings, we have removed the last sentence from the abstract (line 18) and we have clarified it in section 4 (lines 447-451): "This study provides a continental-scale analysis of the changes in flood quantiles that have occurred across Europe over five decades, however further research is needed to formally attribute the resulting regional change patterns to potential driving processes".

**Minor comments**

*The paper makes frequent use of the return period terminology. This is conceptually awkward in non-stationary conditions. I would suggest that the authors at least acknowledge this, and perhaps refer to some of the literature on alternative ways of expressing flood rarity.*

We understand that the use of the return period terminology in a non-stationary context may sound ambiguous to some readers. In the manuscript we refer to the return period (rather than to the annual exceedance probability) because, in the engineering practice, it is widely understood what a 100-year flood is. Examples of return period terminology used in a similar non-stationary context in the literature are Renard et al. (2006), Machado et al. (2015), Šraj et al. (2016). For these reasons we prefer to maintain the return period terminology in the manuscript. However, we have clarified the terminology used in lines 123-124: "We investigate changes in flood quantiles associated with fixed annual exceedance probability 1-p, or, equivalently, with fixed return period T=1/(1-p)".

*The Gumbel parameters are modelled as varying with time according to a log-linear relationship. Perhaps the authors could comment on any alternative ways they considered of modelling trend, such as other mathematical forms of the relationship with time, or inclusion of physical covariates in an attempt to improve the identification of the time trends.*

Thank you for pointing this out. We have introduced additional text about alternative ways of modelling trends in section 4 (lines 352-354): "The two parameters are modelled as varying in time according to log-linear relationships. Other relationships with time could be investigated as well as the use of physical covariates.". The use of physical covariates in order to attribute the trends in flood quantiles is actually planned in the next phase of our research.

*The meaning of gamma and S in the equations around lines 103-4 was not clear to me.*

Thank you for spotting this lack of clarity in these lines. The gammas are parameters of the model that control the scaling with catchment area and S is catchment area. We have specified the meaning of S in line 112 and of gamma in line 118.

*The assumption of homogeneity of the windows (section 2.3) seemed to me to need some justification.*

Thank you for rising this point; we have better explained it in section 2.3 (lines 180-181): "The rationale behind the homogeneity assumption is that the spatial windows, given their size, are characterized by comparatively homogeneous climatic conditions (and hence flood generation processes) and processes driving flood changes". Additionally, we discuss the homogeneity assumption in section 4 (lines 387-395): "We have not assessed the statistical homogeneity of the regions in terms of the flood change model used here. One reason is that formal procedures to assess the regional homogeneity, such as for example those used in regional flood frequency analysis (e.g. Hosking and Wallis, 1993; Viglione et al., 2007), are not available at the moment. Also, while deviation from regional homogeneity would probably invalidate estimates of local flood change statistics from the regional information (e.g., as in the prediction in ungauged basins, see Blöschl et al., 2013b), we expect

its effect on the average regional behavior to be less relevant. This is because we have not observed significant differences in the spatial change pattern when changing the size of the moving windows (not shown here). As a limiting case, the results obtained using the three macro-regions (Sect. 3.3) are consistent with those obtained by the moving window analysis across Europe (Sect. 3.2)."

*I was impressed with the design of Figs 2 and 3, which pack in a great deal of information. I would suggest that the authors either remove or justify the extrapolation of the model to catchment areas ten times smaller and ten times larger than those included in the dataset.*

We agree with the Referee's suggestion of removing the extrapolation of the model to very small and very large catchment areas. We have modified Figures 2 and 3 accordingly.

*The description of Fig 5 mentions larger positive trends in NW France for big floods than for small floods. I could not see that effect from comparing the pairs of maps.*

Thank you for noticing it; we have corrected the description of Figure 5 by removing "and north-western France" from line 268 and 272.
* * *
**Anonymous Referee #3:**

*The article is very nice, and contains a lot of information and results. One thing which is not clear from is the catchment size. For example, the Rijn has a catchment size of 180.000 km2, and contains also smaller catchments. How is this handled in this paper? Can smaller catchments be part of larger catchments? This is important, because large catchments do show a negative trend. This is not explained, and maybe there is no explanation, but has to be investigated in the future, this , however, can be stated more explicitly. The following citation does NOT explain why the large catchment show different results: "Furthermore, in medium and large catchments the magnitude of the trends is in general smaller compared to the small catchments. This may be due to long-duration synoptic weather events, producing floods in medium and large catchments, in contrast to small catchments in western Europe where the largest peaks are often caused by summer convective events with high local intensities". Does this suggest that "long-duration synoptic weather events" do show a negative trend?*

We thank the Anonymous Referee #3 for the time she/he has spent on our manuscript and for the useful and constructive comments.

In this study, we analyse flood data from 2370 hydrometric stations in Europe, each one corresponding to a specific catchment size/area. Consequently, multiple smaller catchments can be part of a larger catchment. We have clarified it in lines 158-160: "Their contributing catchment area ranges from 5 to 100 000 $km^2$ and several nested catchments are included in the database". As explained in section 2.3, the Gumbel regional model is fitted to flood records that are pooled over regions. In each region, the regional trend corresponding to a large (hypothetical) catchment area is an output of the model and is determined by the flood records, within the region, that correspond to large catchment areas. The trend in large catchments is shown in Figure 5b and d and its sign and estimated magnitude varies according to the location and the flood quantile considered.

In this work we do not aim at investigating the causes of the observed flood trends (this is actually planned for the next phase of this research). However, in the discussion section, we make hypotheses on the possible drivers of the resulting flood trends, based on the literature and on our understanding of these processes. The sentences pointed out by the Referee refer to the Atlantic region, where large catchments exhibit, in general, smaller trends (positive for the 2-year quantile and negative for the 100-year quantile) compared to the smaller catchments. We do not intend to attribute this difference to specific drivers, nor to suggest negative trends in long-duration synoptic weather events. These aspects will be investigated in future work. In these lines we hypothesize that different types of weather events could affect the flood trends in catchments of different size in different ways. We have changed these sentence (lines 407-411) from: "This may be due to long-duration synoptic weather events, producing floods in medium and large catchments, in contrast to small catchments

in western Europe where the largest peaks are often caused by summer convective events with high local intensities" to: "This could be explained by different types of weather events and their changes affecting the flood trends in catchments of different sizes in different ways, for example, long-duration synoptic weather events are probably more influential in producing floods in medium and large catchments, in contrast to small catchments in western Europe where the largest peaks are often caused by summer convective events with high local intensities […]".
* * *

[revised manuscript text omitted]